# A Dynamic Constitutive Model and Simulation of Braided CFRP under High-Speed Tensile Loading

**DOI:** 10.3390/ma15186389

**Published:** 2022-09-14

**Authors:** Wei Jin, Yingchuan Zhang, Lanxin Jiang, Guangwu Yang, Jingsong Chen, Penghang Li

**Affiliations:** 1Locomotive and Car Research Institute, China Academy of Railway Sciences Co., Ltd., Beijing 100081, China; 2Beijing Zongheng Electro-Mechanical Technology Co., Ltd., Beijing 100094, China; 3State Key Laboratory of Traction Power, Southwest Jiaotong University, Chengdu 610031, China

**Keywords:** woven CFRP, finite element simulation, dynamic constitutive model, strain rate effect

## Abstract

In this study, a dynamic constitutive model for woven-carbon-fiber-reinforced plastics (CFRP) is formulated by combining dynamic tensile test data and fitting curves and incorporating variation rules established for the modulus of elasticity, strength, and fracture strain with respect to the strain rate. The dynamic constitutive model is then implemented with finite element software. The accuracy and applicability of the dynamic constitutive model are evaluated by comparing the numerically predicted load–displacement curves and strain distributions with the test data. The stress distribution, failure factor, modulus, and strength of the material under dynamic tension are also explored. The results show that the response simulated with the dynamic constitutive model is in good agreement with the experimental results. The strain is uniformly distributed during the elastic phase compared with the DIC strain field. Subsequently, it becomes nonuniform when stress exceeds 600 MPa. Then, the brittle fracture occurs. With the increase in the strain rate, the input modulus decreased, and the tensile strength increased. When the displacement was 0.13 mm, the simulation model was damaged at a low strain rate, and the stress value was 837.8 MPa. When it reached the high strain rate of 800 s^−^^1^, no failure occurred, and the maximum stress value was 432.5 MPa. For the same specimen, the strain rate was the smallest on both clamped ends, and the modulus and strength were large at the ends and small in the middle. The fitting curve derived from the test data was completely input into the dynamic constitutive model to better capture the dynamic change in the material properties.

## 1. Introduction

Carbon-fiber-reinforced composites are characterized by their light weight and high strength and were first used in the aerospace field. With the advances in science and technology, they have been gradually adopted in other fields such as rail transportation. Irrespective of whether it is an aircraft or a rail vehicle, collisions may occur, causing deformation and damage to the material. Therefore, studying the mechanical properties of materials under high strain rates can improve the reliability of structures and help design passive safety and crashworthiness measures.

The dynamic mechanical response of fiber-reinforced composites is typically investigated through experiments [1,2,3,4]. Al-Zubaidy [5] performed dynamic tensile tests on CFRP-bonded joints, finding that their modulus, tensile strength, fracture strain, and energy absorption varied as a function of the strain rate, and establishing corresponding fitted curves for these parameters with respect to the strain rate. Naik [6] performed tensile tests at strain rates of 140–400 s^−1^ and found that the tensile strength of a woven E-glass/epoxy composite increased by 75–93% compared to its value under quasi-static conditions, and that the material with a plain weave had almost the same properties in both the warp and weft directions. Naresh [7] compared the dynamic tensile properties of CFRP, GFRP, and hybrid laminates at strain rates between 0.0016 s^−1^ and 542 s^−1^, observing that GFRP was the more sensitive to changes in the strain rate, with its strength increasing by 66.3%, while those of CFRP and hybrid laminate increased only 6.3% and 39%, respectively. Zhang [8] found that the effect of the strain rate on the behavior of CFRP should be considered differently for unidirectional sheets and fabrics.

The strain rate effect is more pronounced for plain-woven CFRP than for unidirectional CFRP because of the restraint provided by the fibers in the warp direction, and the failure modes of both types are different. Weng [9] investigated the dynamic shear properties of CFRP and found that the in-plane shear characteristics are more sensitive to the strain rate than the interlaminar shear, and the empirical formulas for the strength and modulus of CFRP at high strain rates were determined. After an extensive literature review, Ahmed [10] concluded that the resin material in FRP is more sensitive to the strain rate and plays a more dominant role in the change in strength and modulus. When a load is applied in the fiber direction, the fiber fracture is dominant at low strain rates, and the fiber pull-out is dominant at high strain rates.

A large amount of experimental data helps establish constitutive equations for CFRP under different strain rates [11,12,13], to simulate and analyze its dynamic performance. The methods of FRP simulation are usually divided into micro- and macro-level modeling. Sato [14] formulated a finite element (FE) model of the fiber and matrix, considering interfacial failure and continuum damage mechanics. In addition, he added the effect of temperature on the response of CFRP while considering the strain rate dependency, which enabled his model to predict the damage initiation point and failure mode. Koyanagi [15] applied the FE^2^ multi-scale simulation method to establish a microscopic model and applied it to a CFRP circular tube structure, effectively predicting the initial damage and failure process of a structure under transverse compression. Sawamura [16] used a multi-scale analysis method to consider the properties at the interface between the fiber and resin in the microscopic model and the strain rate effect of the resin, combined with experimental data to obtain the transverse failure envelope and the fiber axial shear strength. Sawamura determined the fiber axial tensile strength and compressive strength using the fiber failure model and the microbuckling model and applied them to the macroscopic FE model.

Al-Zubaidy [17] performed dynamic simulations of CFRP-glued joints, achieving good agreement between the simulation and experimental results. Batuwitage [18] formulated an FE model to evaluate the performance of carbon fiber hollow square tubes under axial impact loading. The effects of the impact mass, impact velocity, bond strength, and fiber modulus on the impact performance of the tubes were investigated, noticing that the variation in impact velocity and fiber modulus had significant effects on the impact response of the tubes. Zhu [19] explored the crushing of CFRP tubes under quasi-static and dynamic loading conditions and found that the loading rate had little effect on the deformation of the composite tubes. The energy absorption capacity under dynamic loading was significantly lower than that under quasi-static loading, but the changes in the modulus and strength of the CFRP under high strain rates were not considered. After conducting dynamic tensile tests of FRP, Kim et al. [20,21] synthesized the variation laws for the material properties with respect to the strain rate and applied them to simulations.

From the aforementioned research, it can be concluded that the existing CFRP dynamic constitutive models do not fully consider the variation law of the modulus, strength, and fracture strain with respect to the strain rate, and they mostly apply to unidirectional plates, while no dynamic constitutive model yet exists for woven CFRP. Taking such limitations into account, this study proposes a dynamic constitutive model for woven CFRP that incorporates the changes in its modulus, strength, and fracture strain as a function of the change in the strain rate using a fitting formula provided in the literature [22]. The dynamic constitutive model was implemented in a finite element software to analyze the strain, stress, and failure factor distribution in CFRP under dynamic tensile loading, as well as the modulus and strength distribution under different strain rates. This study provides a parametric analysis of more laying modes, strain rates, and structural thicknesses, and an in-depth analysis of the dynamic mechanical properties of CFRP, thus providing valuable reference for future engineering applications.

## 2. Dynamic Constitutive Model

### 2.1. Traditional Composite Constitutive Model

In the traditional constitutive model of orthotropic materials, the flexibility matrix is [23]:(1)$$S=\left[\begin{array}{cccccc} \frac{1}{E_{11}} & -\frac{\nu_{12}}{E_{22}} & -\frac{\nu_{13}}{E_{33}} & 0 & 0 & 0\\ -\frac{\nu_{21}}{E_{11}} & \frac{1}{E_{22}} & -\frac{\nu_{23}}{E_{33}} & 0 & 0 & 0\\ -\frac{\nu_{31}}{E_{11}} & -\frac{\nu_{32}}{E_{22}} & \frac{1}{E_{33}} & 0 & 0 & 0\\ 0 & 0 & 0 & \frac{1}{G_{23}} & 0 & 0\\ 0 & 0 & 0 & 0 & \frac{1}{G_{31}} & 0\\ 0 & 0 & 0 & 0 & 0 & \frac{1}{G_{12}} \end{array}\right]$$
where $E_{1}$, $E_{2}$, and $E_{3}$ are the elastic moduli of the material in the three main elastic directions; $G_{12}$, $G_{13}$, and $G_{23}$ are the elastic shear moduli in the three planes; and $\nu_{ij}$ is the Poisson’s ratio relating the strain in the i direction to the strain in the j direction, which satisfies the relation: $\nu_{ij}/E_{j}=\nu_{ji}/E_{i}$ (i,j=1,2,3 but i≠j).

In finite element software, the stiffness matrix is usually used for the iterative calculation of cell’s stress–strain, and a cumulative reduction in stiffness is also applied during the material failure stage; therefore, the flexibility matrix shall be transformed into the stiffness matrix, **C** inverting the flexibility matrix, that is, $S^{-1}=C$. When carbon fiber composites are loaded in different directions, the stress–strain state of the material must also be rotated using the three-dimensional stress transformation formula [20], as follows:(2)$$\bar{C}=T_{\varepsilon }CT_{\varepsilon }^{T}$$
(3)$$T_{\varepsilon }=\left[\begin{array}{cccccc} m^{2} & n^{2} & 0 & 0 & 0 & 2mn\\ n^{2} & m^{2} & 0 & 0 & 0 & -2mn\\ 0 & 0 & 1 & 0 & 0 & 0\\ 0 & 0 & 0 & m & n & 0\\ 0 & 0 & 0 & -n & m & 0\\ mn & -mn & 0 & 0 & 0 & m^{2}-n^{2} \end{array}\right]$$
where m=cosθ, n=sinθ, $\bar{C}$ is the transformed stiffness matrix, $T_{\varepsilon }$ is the stress transformation matrix, and θ is the angle of fiber rotation.

Table 1 presents the Hashin three-dimensional failure criterion, which is typically used to determine material failure [21]. It can be seen that the delamination failure is also considered in the Hashin failure criterion, i.e., matrix tensile delamination failure and matrix compressive delamination failure.

In Table 1, $X_{T}$, $X_{C}$, $Y_{T}$, $Y_{C}$, $Z_{T}$, $Z_{C}$, $S_{12}$, $S_{23}$, and $S_{13}$ represent the tensile strength of the material in the X-direction, compressive strength in the X-direction, tensile strength in the Y-direction, compressive strength in the Y-direction, and shear strength in the three directions, respectively.

The stiffness reduction coefficients for each failure mode are typically adopted from the scheme proposed by Alvaro [21], as shown in Table 2. When delamination failure occurs in the simulation process, the values of $E_{3}$, $G_{13}$ and $G_{23}$ will become 0, and the material will not be able to bear in the vertical direction.

### 2.2. Modified Dynamic Constitutive Model

The dynamic constitutive model used in this study incorporates the variation in strength, modulus, and fracture strain with respect to the strain rate, based on data obtained from the literature [22]. The tensile process was conducted on a Zwick/Roell HTM5020 high-speed tensile tester (Zwick/Roell Company, Ulm, Germany). Failure of the fiber and matrix at the fracture was observed via SEM. The stress–strain curves of the tensile test, the strain fields recorded by DIC, the failure modes of the specimens, and the dynamic fitting formula by data analysis were also from the literature. In addition, woven CFRP specimens were subjected to tensile tests using six strain rates between 1 and 800 s^−1^ and laying modes of [0/90]_12_, [0/90/±45]_3s_, and [±45]_12_, as shown in Figure 1. These three layups were consistent with those in the literature at the time of the tests. The above works were used as the input data of the simulation and the reference of the calculation results in this paper. The empirical formulation was derived by fitting a large amount of test data. By extracting the ultimate stress values from the stress–strain curves at the moment of the fracture, the function of strength relative to the strain rate could be obtained by fitting. The curves of the elastic modulus related to the strain rate were sorted as an approximate quadratic function. 

In the dynamic constitutive model, the modulus and strength at [0/90]_12_ were selected as the input values for $E_{1}$, $E_{2}$, $X_{T}$ and $Y_{T}$, and the modulus and strength at [±45]_12_ were selected as the input values for $G_{12}$ and $S_{12}$, respectively.

The average effective strain rate of each element is then calculated as the current element strain rate value and matched to the dynamically updated values of the material’s modulus and strength under different strain rates obtained from the tests. Owing to the characteristics of two-dimensional woven composites, $E_{11}=E_{22}$ and $X_{T}=Y_{T}$ in the constitutive model. When updating the modulus and strength values under dynamic conditions, the effective strain rate of the element is first computed as follows [24]:(11)$$\dot{\bar{\varepsilon }}={\left(\frac{2}{3}{\dot{\varepsilon }}_{ij}{\dot{\varepsilon }}_{ij}\right)}^{\frac{1}{2}}$$

Applying the Yen and Caiazzo (Y-C) formula [7,25,26]:(12)$$y=\left(x_{0}\varphi \ln\frac{\dot{\bar{\varepsilon }}}{{\dot{\varepsilon }}_{0}}+x_{0}\right)$$
where y represents the modulus and strength values affected by the strain rate, and $x_{0}$ represents the initial modulus and strength values at state ${\dot{\varepsilon }}_{0}$.

After considering the modulus and strength change rules, the fracture strain at different strain rates is also considered and substituted into the equations describing the relationship between the fracture strain and strain rate for [0/90]_12_ and [±45]_12_. The final inputs in the dynamic constitutive model are [22]:(13)$$E_{11}=E_{22}=E_{1\mathrm{dynamic}}=63746.6-22.46\dot{\varepsilon }$$
(14)$$X_{T}=Y_{T}=X_{\mathrm{Tdynamic}}=54.1{\left(\mathrm{lg}\dot{\varepsilon }\right)}^{2}-85.5\mathrm{lg}\dot{\varepsilon }+616.8$$
(15)$$S_{12\mathrm{dynamic}}=\left\{\begin{array}{l} 107.7+5.77\mathrm{lg}\dot{\varepsilon },\mathrm{lg}\dot{\varepsilon }\leq 2.4\\ -25.2+60.7\mathrm{lg}\dot{\varepsilon },\mathrm{lg}\dot{\varepsilon }>2.4 \end{array}\right.$$
(16)$$\varepsilon_{1\mathrm{dynamic}}=0.00962\dot{\varepsilon }+3.31\times 10^{-6}$$
(17)$$\varepsilon_{12\mathrm{dynamic}}=\left\{\begin{array}{l} 0.092-10^{-4}\dot{\varepsilon },1<\dot{\varepsilon }\leq 250\\ 0.068-4.73\times 10^{-6}\dot{\varepsilon },250<\dot{\varepsilon }<800 \end{array}\right.$$

From the above equations and for strain rates between 1 to 800 s^−1^, as the strain rate increases, the modulus of elasticity decreases, whereas the ultimate tensile strength increases. Furthermore, when the strain value of the [0/90]_12_ ply element is greater than that of $\varepsilon_{1\mathrm{dynamic}}$ or the strain value of the [±45]_12_ ply element is greater than that of $\varepsilon_{12\mathrm{dynamic}}$, complete failure of the material occurs.

## 3. Finite Element Model

### 3.1. UMAT

The simulations run in this study were completed using the FE software LSDYNA (Livermore Software Technology Corporation, Livermore, CA, USA). The material database included in the software does not currently include a constitutive model for carbon fiber composites under dynamic effects; therefore, it is necessary to write such a material subroutine. The dynamic constitutive model of the material was formulated based on experimental data, as discussed in Section 2. The material subroutine must be written using file 21.F in the user-material package. A self-defined material constitutive equation is then written using UMAT 41–50. Subsequently, a new lsdyna.exe solver is generated by the Fortran compiler (Microsoft, Redmond, WA, USA), and a K input file is submitted for solving. A flowchart of the subroutine calculation is shown in Figure 2. First, the load is applied, and the modulus and strength values are calculated based on the average strain rate of the element; then, the Jacobian matrix is computed to determine the stress and failure factors. If the model does not meet the failure criteria, the element stress and strain values are directly output. Otherwise, stiffness degradation is performed, and the Jacobian matrix is recomputed to obtain corrected stress and strain values.

When either the tensile coefficient or the strain value of the fiber element is greater than the fracture strain, the failure criteria are considered as met and the element is deleted. When writing UMAT, the history variable hsv(i) is set to store the values of the seven failure factors in Table 1, as well as the values of the strain rate and modulus for each element, enabling the display of contour maps for easy visual comparison and analysis at a later stage. The material subroutine we wrote did not consider the inertia problem, but only focused on the stress–strain relationship of the material. As a dynamic simulation software, LSDYNA has its own algorithm. Inertia has been considered in the deformation law of the element under high-speed and low-speed conditions.

### 3.2. Dynamic Tensile FE Model

The use of thick-shell (TShell) elements to simulate composite connecting plates better captures the local and global effects and yields more precise results [24]. The test coupon model was divided into six layers of elements across the thickness direction, each layer was set at four angles, each element was defined by four integrations points, and the resultant output values were recorded for each composite layer. The composite plate was assigned the proposed material model, which can capture the damage to the composite fiber and matrix during the tensile process. The elements have an approximate size of 1 mm, and the mesh is refined to an element size of 0.5 mm in the middle section.

The resulting finite element model of the specimen used in the simulation of the dynamic tensile tests, which comprises 13,620 elements and 16,985 nodes, is shown in Figure 3. According to the depth of the fixture during the test, the right end was constrained, and a uniform velocity was applied to the left end. The total calculation time was 0.01 s, and the loading velocity was computed as $v=\dot{\varepsilon }L_{\mathrm{gaugelength}}$, where $\dot{\varepsilon }$ is the strain rate and $L_{\mathrm{gaugelength}}$ is the gauge length of the working section of the test piece.

The material parameters used in the simulations are listed in Table 3. The initial modulus and strength data were obtained via tensile and compression tests under quasi-static conditions, where the strain rate $\dot{\varepsilon }$ was approximately equal to 0.

## 4. Analysis of Simulation Results

### 4.1. Comparison of Experimental and Simulation Results

The dynamic constitutive model proposed in this study was developed by fitting the data from the tests conducted at [0/90]_12_ and [±45]_12_. Therefore, in the comparison of results presented in this section, the data from the tests with a mixed-angle lay-up, i.e., [0/90/±45]_3s_, were used. Figure 4 shows the experimental and numerical stress–strain curves of the specimens subjected to tensile loading at strain rates of 1, 250, and 800 s^−1^. The stress and strain were taken as the average values along the gauge length in the middle section. It is observed that as the strain rate increases, the tensile strength of the specimen increases, with the simulation results being in good agreement with the test data. The stress–strain curve is relatively smooth at a low strain rate but becomes noisy as the strain rate increases.

During the tests, the damage process and the strain variation along the gauge length of the specimens were recorded using the digital image correlation (DIC) technique. Figure 5 compares the simulated transverse strain fields *ε**_xx_* for the specimen tested with a [0/90/±45]_3s_ lay-up at a strain rate of 250 s^−1^ to those recorded using DIC, with purple representing a low strain value of 0.006, and red corresponding to a high strain of 0.014. Each picture corresponds to a specific stress–strain value pair, as labeled at the bottom of the image. The stresses were determined from the stress–strain curve based on the strain value. An assessment of the damage evolution demonstrates that the strain distribution was relatively uniform across the specimen during the linear elastic stage, subsequently turning non-uniform as the load increased and the strain reached a value of 0.012–0.013. Owing to the presence of random imperfections, carbon fiber bundles with varying stiffness and potentially eccentric loads can cause nonuniformity. The load at this stage was close to the limit value, and a local stress concentration appeared in the strain field (red area). The fracture of the specimen eventually occurs at this location, leading to an abrupt stress drop. The simulated and experimental strain distributions were in good agreement, indicating that the proposed dynamic constitutive finite element model can satisfactorily reproduce the strain change and damage process of the specimen.

### 4.2. Stress and Failure Contour Maps

Figure 6 shows the stress contour maps of specimens tested with a lay-up angle of [0/90/±45]_3s_ at different strain rates, when the displacement reaches 0.13 mm. At a strain rate of 10 s^−1^, some elements in the gauge length had failed and been deleted, the specimen was about to fracture, and the maximum stress was 837 MPa. At a strain rate of 250 s^−1^, none of the elements had failed, and the maximum stress value was 667.8 MPa. At a strain rate of 800 s^−1^, due to the strain rate effect, the modulus decreased, the tensile strength increased, the stress decreased under the same displacement, and the maximum stress reached 432.5 MPa.

Figure 7 shows the contour map of the fiber tensile failure factor for the three different lay-ups considered ([0/90]_12_, [0/90/±45]_3s_, and [±45]_12_) at a displacement of 0.15 mm and a strain rate of 250 s^−1^. The failure mode of the specimen under tension mainly consisted of fiber tensile fracture; therefore, the focus of the analysis was on the distribution of the fiber tensile failure factor $F_{t}$. The [0/90]_12_ lay-up specimen was damaged, owing to its large modulus, at a maximum stress of 802.2 MPa, when elements with an $F_{t}$ greater than two were removed. The maximum value of $F_{t}$ in the unremoved elements was 1.917, and the failure mode consisted of a fiber fracture, which agrees with the failure mode observed in the experimental tests. As shown in Figure 8, the modulus of the specimens with ±45 angle plies was reduced, and they were able to withstand greater deformations without being damaged. In the case of [0/90/±45]_3s_, the maximum stress was 610.9 MPa, and the maximum value of $F_{t}$ was 0.7737, without failure and stiffness reduction. With an increase in the angle of the [±45]_12_ lay-up, the maximum stress and $F_{t}$ value were further reduced to 194.1 MPa and 0.3235, respectively.

### 4.3. Analysis of the Strain Rate Effect on the Modulus and Strength

According to the material subroutine written in UMAT, each element has its own effective strain rate value, which, in turn, corresponds to a certain modulus and strength. As shown in Figure 9a, the strain rate gradually increased from 0 s^−1^ at the clamped end to 764 s^−1^ at the gauge length section. The modulus’s distribution is caused by the change in the strain rate. As shown in Figure 9b, the modulus at the clamped end was 64,000 MPa and decreased to 50,225 MPa at the location of the maximum strain rate. In the strength distribution diagram in Figure 9c, because the strength is a quadratic function of $lg\dot{\varepsilon }$, as the strain rate increases, the strength first decreases and then increases. At the clamped end, where the strain rate is 0 s^−1^, the initial value of the strength is 771.8 MPa. In the transition region, the strength decreases to 662.5 MPa. The maximum strength of 821.2 MPa occurred in the middle section, where the strain rate was the highest. Therefore, using this dynamic constitutive model, the distribution of the strain rate, modulus, and strength at each load step, as well as the progress of the dynamic tensile process of CFRP, can be accurately predicted based on the elemental strain rates.

### 4.4. Additional Parametric Analysis

After the numerical results obtained with the dynamic constitutive model proposed in this study were compared with the experimental data, the dynamic tensile response of the material under different strain rates was further investigated. This way, parametric analyses of the dynamic constitutive model can be conducted to comprehensively explore the CFRP dynamic tensile rule. As shown in Table 4, the stress–strain curves in Figure 10 were obtained by simulating dynamic tensile tests of models with five lay-up angles, five strain rates, and five thicknesses.

As shown in Figure 10a, setting a strain rate of 250 s^−1^ and a thickness of 2.4 mm, the response of five types of lay-ups was evaluated. It can be seen from the results that when the percentage of 0/90 angle lay-ups increased, the tensile strength of the material and the slope of the curve increased. When the percentage of ±45 angle lay-ups increased, the curve tended to fluctuate. Figure 10b shows the stress–strain curves for a lay-up of [0/90]_12_ and a thickness of 2.4 mm, using five different strain rates. The general trend showed that as the strain rate increases, the slope of the curve decreases, and the strength of the material increases, which is consistent with the input fitting function. In addition, it also can be seen that the curves in Figure 10b were smoother than those in Figure 4. As shown in Figure 10c, when the strain rate was set to 250 s^−1^ and the lay-up to [0/90]_12_, while the thickness of the model was changed, the results demonstrated that the thickness had little effect on the mechanical response of the material.

## 5. Conclusions

A dynamic constitutive model for woven CFRP was formulated by considering the variation rule of the modulus, strength, and fracture strain under high-speed tensile dynamic loading. The dynamic constitutive model was implemented in LSDYNA to perform numerical simulations. After comparing the numerical and experimental results, the following conclusions can be drawn:

The results simulated with the dynamic constitutive model were in good agreement with the stress–strain curves and strain distributions from the tests during different stages. In the linear elastic stage of the test, the strain distributions of the specimens were relatively uniform. When the strain value reached 0.012–0.013, the strain field turned non-uniform, and failure occurred in the form of a brittle fracture. For the [0/90/±45]_3s_ lay-up, as the strain rate increased, the input modulus decreased, the tensile strength decreased, and the stress decreased under a constant displacement. Comparing the three lay-ups considered, the [0/90]_12_ lay-up resulted in the highest stress for a given displacement, with failure mainly consisting of fiber tensile fracture.For the same specimen, the strain rate was the lowest at both clamped ends and the largest at the middle section. On the contrary, the modulus was high on both ends and small in the middle. On the other hand, the strength was the smallest at the clamped ends, decreasing at the transition region and increasing again toward the middle section where the strain rate was high. Combined with the simulation of additional lay-ups, strain rates, and thicknesses, it was found that the fitting curve in the test is completely input into the dynamic constitutive model, which can better reflect the dynamic variation rules of the different material properties. It was also found that the tensile strength of the material and the slope of the curve increased as the percentage of 0/90 angle lay-ups increased, while as the percentage of ±45 angle lay-ups increased, the curve tended to fluctuate. As the strain rate increased, the slope of the curve decreased, and the strength of the material increased.The dynamic properties of woven CFRP have not yet been extensively studied. Different resins, fibers, and weaving methods can affect the dynamic properties of the material. In the field of rail transportation, the application of composite materials is gradually increasing. Research on dynamic constitutive models of composite materials enables a better simulation of the bearing capacity and energy absorption characteristics of the structure during train collisions, which may significantly promote the application of carbon fiber composites in other multiple fields.

## Figures and Tables

**Figure 1 materials-15-06389-f001:**
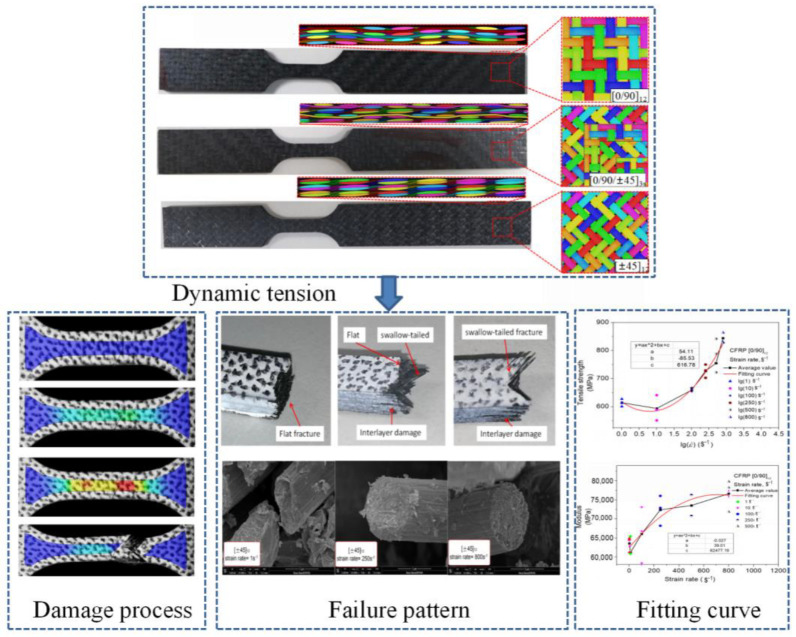
Dynamic tensile test of composite [22].

**Figure 2 materials-15-06389-f002:**
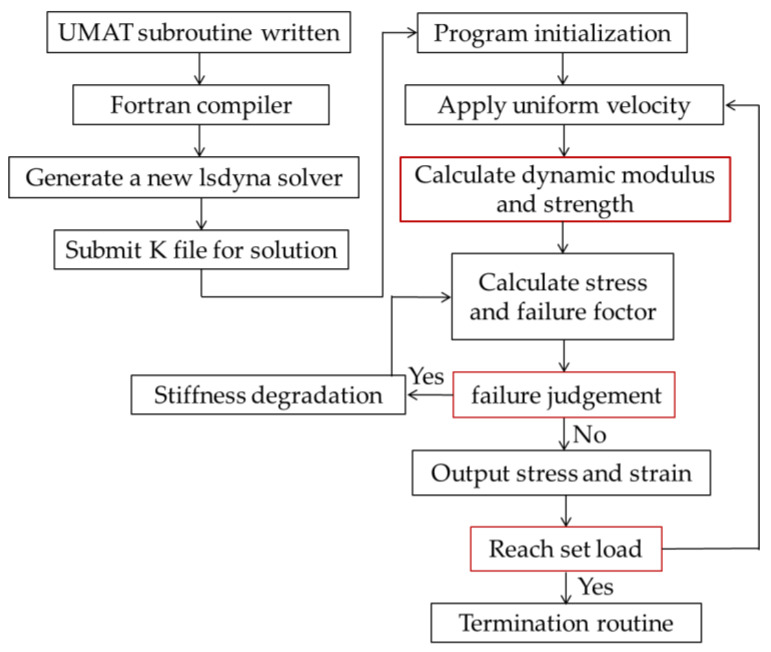
Flowchart of the subroutine calculation.

**Figure 3 materials-15-06389-f003:**
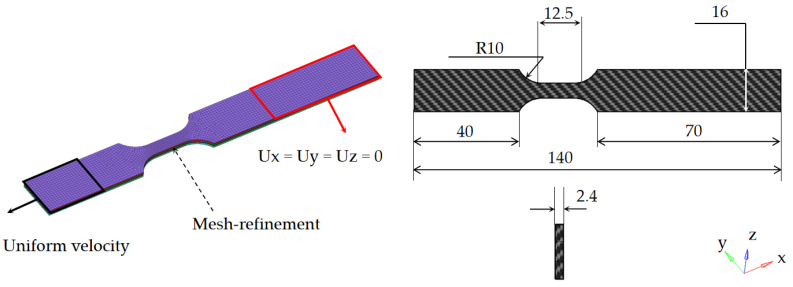
Finite element model.

**Figure 4 materials-15-06389-f004:**
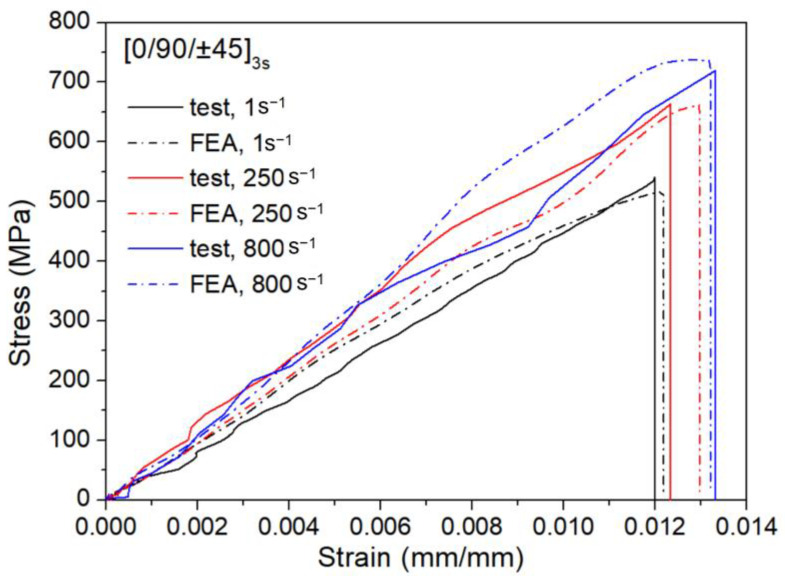
Comparison of simulated and experimental stress–strain curves for [0/90/±45]_3s_.

**Figure 5 materials-15-06389-f005:**
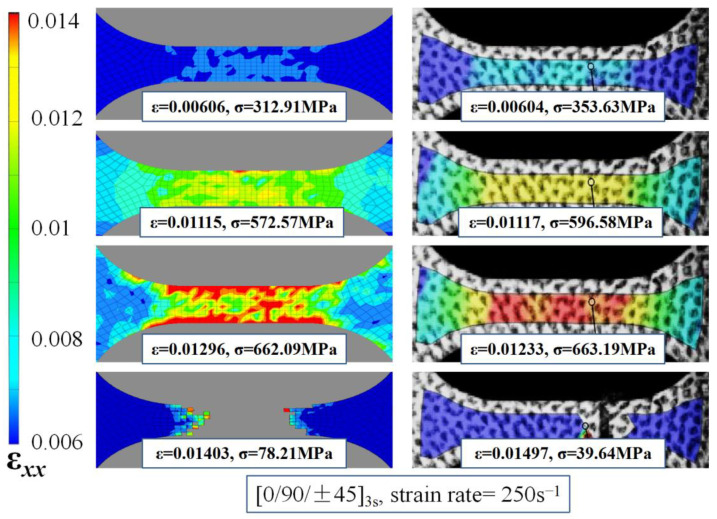
Simulated and DIC strain distributions.

**Figure 6 materials-15-06389-f006:**
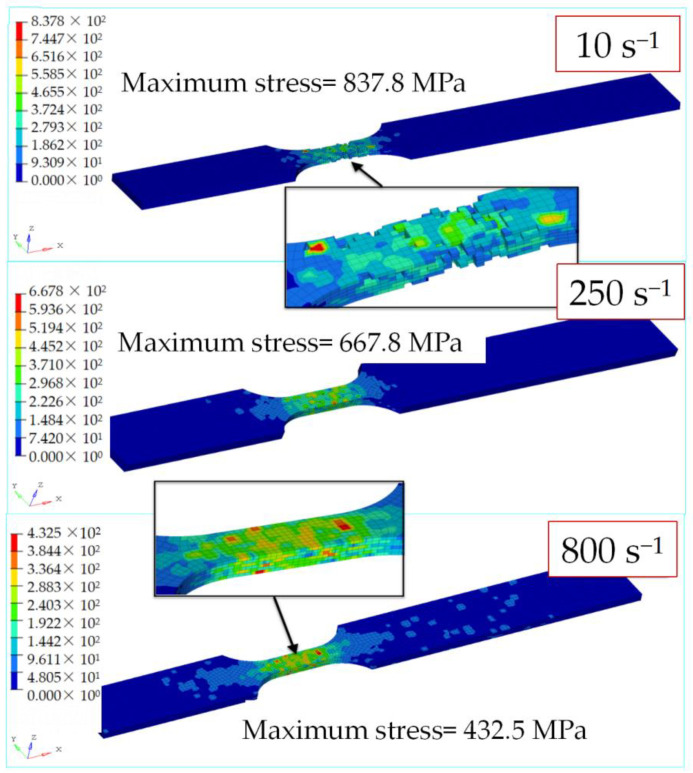
Stress contour map of [0/90/±45]_3s_ at different strain rates when d = 0.13 mm.

**Figure 7 materials-15-06389-f007:**
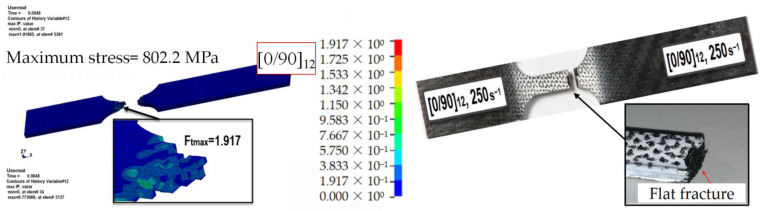
Fiber tensile failure factor contour map and failure mode of the [0/90]_12_ specimen tested at a strain rate of 250 s^−1^ when d = 0.15 mm.

**Figure 8 materials-15-06389-f008:**
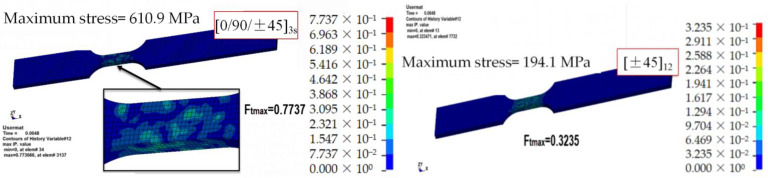
Fiber tensile failure factor contour map of the [0/90/±45]_3s_ and [±45]_12_ specimens tested at a strain rate of 250 s^−1^ when d = 0.15 mm.

**Figure 9 materials-15-06389-f009:**
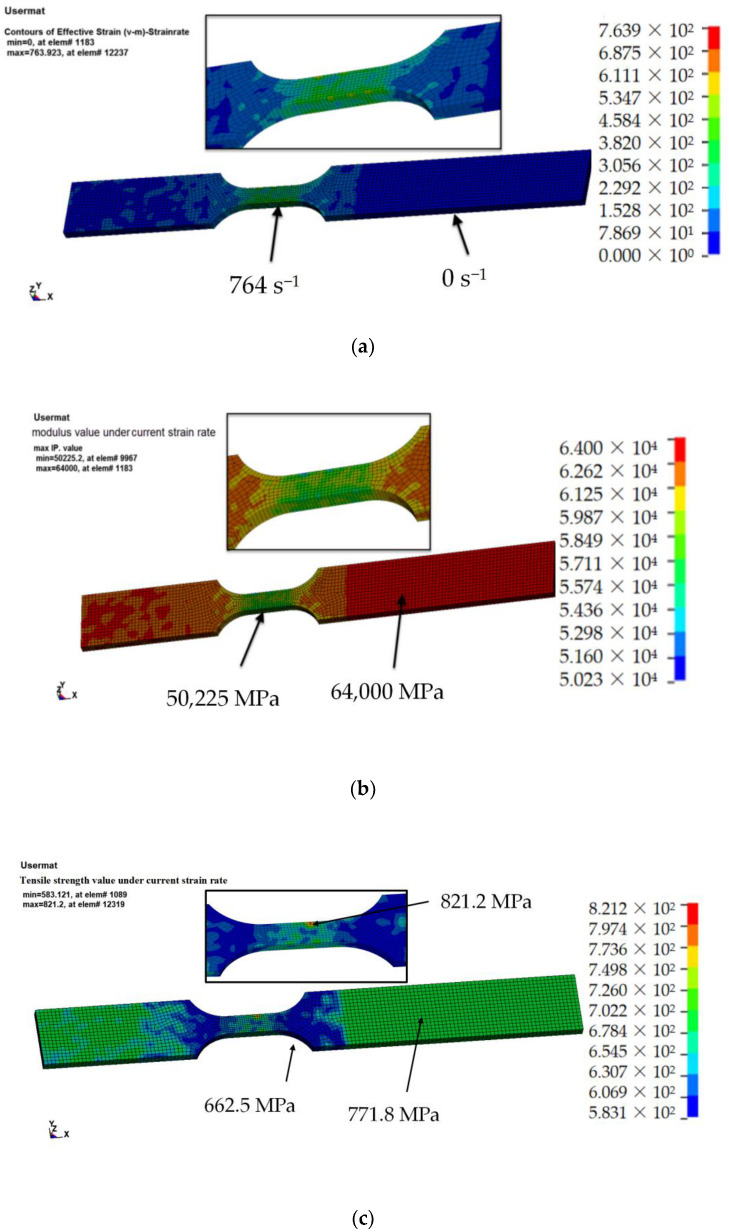
Contour maps showing the distribution of the (**a**) strain rate, (**b**) modulus, and (**c**) strength across the finite element model.

**Figure 10 materials-15-06389-f010:**
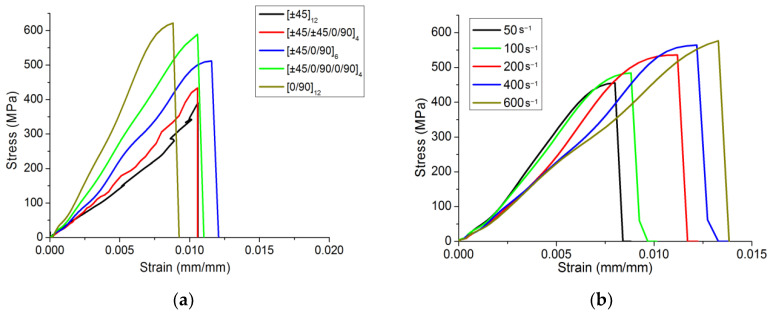
Stress–strain curves obtained with the finite element model for different (**a**) lay-ups, (**b**) strain rates, and (**c**) thicknesses.

**Table 1 materials-15-06389-t001:** Hashin three-dimensional failure criterion.

Failure Mode	Failure Criteria	
Fiber tensile failure	$F_{t}={\left(\frac{\sigma_{11}}{X_{T}}\right)}^{2}+{\left(\frac{\tau_{12}}{S_{12}}\right)}^{2}+{\left(\frac{\tau_{13}}{S_{13}}\right)}^{2}\geq 1$	(4)
Fiber compressive failure	$F_{c}={\left(\frac{\sigma_{11}}{X_{C}}\right)}^{2}\geq 1$	(5)
Matrix tensile failure	$M_{t}={\left(\frac{\sigma_{22}}{Y_{T}}\right)}^{2}+{\left(\frac{\tau_{12}}{S_{12}}\right)}^{2}+{\left(\frac{\tau_{23}}{S_{23}}\right)}^{2}\geq 1$	(6)
Matrix compressive failure	$M_{c}={\left(\frac{\sigma_{22}}{Y_{C}}\right)}^{2}+{\left(\frac{\tau_{12}}{S_{12}}\right)}^{2}+{\left(\frac{\tau_{23}}{S_{23}}\right)}^{2}\geq 1$	(7)
Fiber-matrix shear-out failure	$S={\left(\frac{\sigma_{11}}{X_{C}}\right)}^{2}+{\left(\frac{\tau_{12}}{S_{12}}\right)}^{2}+{\left(\frac{\tau_{13}}{S_{13}}\right)}^{2}\geq 1$	(8)
Matrix tensile delamination failure	$M_{\mathrm{tl}}={\left(\frac{\sigma_{33}}{Z_{T}}\right)}^{2}+{\left(\frac{\tau_{13}}{S_{13}}\right)}^{2}+{\left(\frac{\tau_{23}}{S_{23}}\right)}^{2}\geq 1$	(9)
Matrix compressive delamination failure	$M_{\mathrm{tl}}={\left(\frac{\sigma_{33}}{Z_{C}}\right)}^{2}+{\left(\frac{\tau_{13}}{S_{13}}\right)}^{2}+{\left(\frac{\tau_{23}}{S_{23}}\right)}^{2}\geq 1$	(10)

**Table 2 materials-15-06389-t002:** Material parameter degradation per failure mode.

Failure Mode	$E_{1}$	$E_{2}$	$E_{3}$	$G_{12}$	$G_{13}$	$G_{23}$	$v_{12}$	$v_{13}$	$v_{23}$
Fiber tensile failure	0.14	0.4	0.4	0.25	0.25	0.2	0	0	0
Fiber compressive failure	0.14	0.4	0.4	0.25	0.25	0.2	0	0	0
Matrix tensile failure	-	0.4	0.4	-	-	0.2	0	0	0
Matrix compressive failure	-	0.4	0.4	-	-	0.2	0	0	0
Fiber-matrix shear-out failure	-	-	-	0.25	0.25	-	0	0	-
Matrix tensile delamination failure	-	-	0	-	0	0	-	0	-
Matrix compressive delamination failure	-	-	0	-	0	0	-	0	-

**Table 3 materials-15-06389-t003:** Composite model parameters.

Parameter	Assigned Value	Parameter	Assigned Value
$E_{1}$	64.00 GPa	$X_{T}$	771.83 MPa
$E_{2}$	64.00 GPa	$X_{C}$	830.93 MPa
$E_{3}$	10.30 GPa	$Y_{T}$	771.83 MPa
$G_{12}$	4.97 GPa	$Y_{C}$	830.93 MPa
$G_{13}$	4.97 GPa	$Z_{T}$	31.2 MPa
$G_{23}$	3.50 GPa	$Z_{C}$	184 MPa
$v_{12}$	0.066	$S_{12}$	107.7 MPa
$v_{13}$	0.3	$S_{13}$	94.24 MPa
$v_{23}$	0.3	$S_{23}$	94.24 MPa

**Table 4 materials-15-06389-t004:** Variable design.

Different Lay-Ups	[0/90]_12_	[±45/0/90/0/90]_4_	[±45/0/90]_6_	[±45/±45/0/90]_4_	[±45]_12_
Different strain rates/s^−1^	50	100	200	400	600
Different thicknesses/mm	1.6	2	2.4	2.8	3.2

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
