# Peer review of "A Dynamic Constitutive Model and Simulation of Braided CFRP under High-Speed Tensile Loading"

_materials, 2022, doi:10.3390/ma15186389_

Round 1

Reviewer 1 Report

The authors have trie to demonstrate the applicability of strain rate dependence of constitutive model parameters on the overall behavior of CFRP composites. The current manuscript needs major modifications and explanations before it can be considered for publication. 

1. The authors have mentioned at several places that they have studied ‘dynamic constitutive model’. However the only modification done here is the the use of strain rate dependent constants. ‘Dynamic’ constitutive models I mean that the behavior is also dynamic, i.e., inertia becomes significant. Authors should look at the paper ‘A self-consistent homogenization framework for dynamic mechanical behavior of fiber reinforced composites, Mechanics of Materials, volume 166, 2022, 104222‘ and the references there.

Do the authors here have included inertia terms as well? Statements in the manuscript seems to suggest that the simulations are all done under quasi-static conditions.

2. In Fig. 3, at high strain rates, stress-strain curves have fluctuations in the elastic region in the simulations as well. If there’s no inertia, this doesn’t seem plausible. Could the authors explain the reasons for this behavior?

3. The samples used have no pre-crack to initiate the fracture. In that case it is most likely to delaminate at the ply interfaces. Since the have not included interface delamination in their model, does it mean that the experiment also do not show any such failure of interfaces?

4. What was the rationale behind ignoring interface failure modeling.

5. Comparing Fig. 3 with Fig. 9, at high strain rates, the behavior in Fig. 9 is smooth. Was this solely because of the difference in layup? Could the authors explain the inconsistencies here?

Reviewer 2 Report

Abstract - missing key numerical results.

Please split large paragraphs for ease of reading.

Lots of standard equations are included - are these needed?

Where are the numerical values in 12 to 18 obtained from?

Flowchart is relatively conventional.

Insufficient detail on experimental testing provided. There is next to nothing given - but there should be sufficient detail as to allow an independent researcher to repeat the study.

Several ineffective figure / scale bar too small to read.

There is insufficient discussion / comparison / validation of the models.

Broader implications lacking detail.

Round 2

Reviewer 1 Report

The manuscript now looks much better and should be considered for publication after improving a few minor mistakes in sentences.

For example, the sentence 'The material subroutine we written ...' is not correct. Please check the manuscript for other mistakes as well and correct it.

Reviewer 2 Report

The English language of the corrections is very poor and requires substantial improvement.

Spaces between variables and units

Abstract - English language issues in corrections.

Authors reference a previous work for experimental approach, however further summary is required (not everyone will have access to this).

Text in several figures remains too small to read
